# Carbonation Behavior of Engineered Cementitious Composites under Coupled Sustained Flexural Load and Accelerated Carbonation

**DOI:** 10.3390/ma15186192

**Published:** 2022-09-06

**Authors:** Hongzhi Zhang, Yingxuan Shao, Ning Zhang, Abdullah M. Tawfek, Yanhua Guan, Renjuan Sun, Changjin Tian, Branko Šavija

**Affiliations:** 1School of Qilu Transportation, Shandong University, Jinan 250002, China; 2Suzhou Research Institute, Shandong University, Jinan 215021, China; 3Shandong Hi-Speed Engineering Test Co., Ltd., Jinan 250002, China; 4Bridge and Tunnel Engineering, Sana’s University, Sanaa 12544, Yemen; 5Microlab, Faculty of Civil Engineering and Geosciences, Delft University of Technology, 2628 CN Delft, The Netherlands

**Keywords:** engineered cementitious composites, sustained flexural load, accelerated carbonation, microstructure, microhardness

## Abstract

Engineered cementitious composites (ECCs) belong to a broad class of fibre-reinforced concrete. They incorporate synthetic polyvinyl alcohol (PVA) fibres, cement, fly ash and fine aggregates, and are designed to have a tensile strain capacity typically beyond 3%. This paper presents an investigation on the carbonation behaviour of engineered cementitious composites (ECCs) under coupled sustained flexural load and accelerated carbonation. The carbonation depth under a sustained stress level of 0, 0.075, 0.15, 0.3 and 0.6 relative to flexural strength was measured after 7, 14 and 28 days of accelerated carbonation. Thermogravimetric analysis, mercury intrusion porosimetry and microhardness measurements were carried out to show the coupled influence of sustained flexural load and accelerated carbonation on the changes of the mineral phases, porosity, pore size distribution and microhardness along the carbonation profile. A modified carbonation depth model that can be used to consider the coupled effect of flexural tensile stress and carbonation time was proposed. The results show that an exponential relationship can be observed between stress influence coefficient and flexural tensile stress level in the carbonation depth model of ECC, which is different when using plain concrete. Areas with a higher carbonation degree have greater microhardness, even under a large sustained load level, as the carbonation process refines the pore structure and the fibre bridges the crack effectively.

## 1. Introduction

Engineered cementitious composites (ECCs) belong to a class of short fibre-reinforced composites [1,2]. They are also termed strain-hardening cementitious composites (SHCCs) [3] or ultra-high toughness cementitious composites (UHTCCs) [4]. They exhibit pseudo-strain-hardening and multiple cracking behaviours under tension. The deformation capacity of ECC under tension is about two orders of magnitude higher than that of plain concrete [5]. It has also been shown that the inclusion of fibres leads to a superior fatigue performance of ECC under a relatively high fatigue stress and strain level compared to plain concrete [5]. More promisingly, the tight crack widths of ECCs allow the material to possess a significant ability to recover mechanical properties through autogenous healing in the presence of water [6,7,8,9].

Given its high cost, ECC is generally used in parts of a structure under high static/fatigue stress and strain. ECC has been used as overlay on steel bridge decks [10,11,12]. Great improvements have been made in terms of the load-bearing capacity and fatigue life. Zhang and Li applied ECCs to a pavement overlay system, and showed that ECC can effectively protect the reflective crack from subgrading [13,14]. Regarding reinforced concrete members, ECC can be used as cover of the reinforcement [15,16,17,18]. The crack width under both static and fatigue load can be controlled effectively. It is believed that under such conditions, the crack width can be effectively controlled to under 100 μm [4]. This limits the penetration and diffusion of aqueous solutions containing aggressive ions, such as chlorides and sulphates [19,20,21], thus delaying the onset of reinforcement corrosion and prolonging the service life of the structure.

During the past few decades, numerous efforts have been carried out to understand the durability of ECC in different environments, as this is of great importance to the safe and efficient operation of infrastructures. It has been shown that, compared with plain concrete/mortar, ECC has lower permeability [22] and stronger resistance to high temperature [23], chloride penetration [24,25], sulphate attack [24] and freeze–thaw cycles [26], especially under coupled mechanical loading and environment exposure conditions. However, not a lot is known about the carbonation resistance of the hardened ECC. In fact, most studies focus on the accelerated carbonation curing of such materials [27,28], which is different from the carbonation processes of hardened materials. [29].

In the carbonation process, CO_2_ dissolves in the pore solution and reacts with the calcium-bearing phases, e.g., portlandite and calcium silicate hydrate (C-S-H), forming CaCO_3_. As it decreases the alkalinity of the matrix, the carbonation of the concrete cover plays an important role on the de-passivation and corrosion of the reinforcement. In terms of ECC, in which cement blended with fly ash is used as the binder material, the carbonation is expected to be more severe because the pozzolanic reaction reduces the portlandite content [30]. This leads to the rapid decalcification of C-S-H, which causes carbonation shrinkage [31]. The ingress of CO_2_ into concrete does not occur at a constant rate. This is attributed to the heterogeneous microstructure and redistribution of the pore sizes of the carbonation zone [32,33]. However, in practice, the concrete is subjected to a sustained load, which may cause creeping and microcracks [34]. To date, a preliminary understanding of the effect of precracking on the carbonation of concrete [35,36,37] has been gained. In view of the differences in crack pattern and pore structure between the loaded and unloaded situations, the issue of carbonation under sustained load has received considerable attention [38,39,40].

To this end, the current study presents an investigation on the carbonation behaviour of ECC under coupled sustained flexural load and accelerated carbonation. The carbonation depth under sustained stress level of 0, 0.075, 0.15, 0.3 and 0.6 relative to flexural strength was measured after 7, 14 and 28 days accelerated carbonation. A modified carbonation depth model that considers the coupled effect of flexural tensile stress and carbonation time was proposed. Thermogravimetric analysis (TGA), mercury intrusion porosimetry (MIP) and microhardness measurements [41,42] were carried out to show the coupled influence of sustained flexural load and accelerated carbonation on the changes of the mineral phases, porosity, pore size distribution and microhardness along the carbonation profile.

## 2. Materials and Methods

### 2.1. Materials and Specimens

In the current study, an ECC composition developed at the Shandong University was chosen [11,25] (see Table 1). Ordinary Portland cement 42.5 I and class F fly ash were used as binder materials ([43]). Their chemical compositions provided by the suppliers are presented in Table 2. The quartz sand has a grain size between 125 and 180 μm [44,45]. The PVA fibre was provided by KURARAY CO, LTD. Polycarboxylate ether-based superplasticizer and viscosity modifying admixture (VMA) were added to adjust the fresh properties for casting. The water to binder (*w/b*) ratio was 0.30. Prismatic specimens with the size of 400 mm × 100 mm × 50 mm were prepared and cured in a room with relative humidity of 95% and temperature of 20 ± 2 °C for 28 days. Three specimens were used for the flexural load-bearing capacity determination. Two loading levels, and three sustained loading periods (7, 14 and 28 days) under accelerated carbonation were considered. This led to 6 groups of specimens. For each group, three specimens were tested. Therefore, 18 prismatic specimens were prepared for the coupled sustained flexural load and accelerated carbonation.

### 2.2. Determination of Flexural Load Bearing Capacity

A four-point flexural test was used to determine the flexural load capacity of the hardened ECC. A span length of 300 mm was adopted. It was instrumented by a universal testing machine with a load capacity of 10 kN and run under displacement control with a loading rate of 0.4 mm/min. The midspan deflection was recorded with a linear variable differential transformer (LVDT). The flexural load capacity of each mixture was determined by averaging the ultimate flexural load resulting from 3 specimens. The first cracking strength and the ultimate flexural strength were determined as 7.41 MPa (corresponding to 6.18 kN in load) and 11.41 MPa (corresponding to 9.51 kN in load), respectively. The ultimate flexural strength was used to determine the load values corresponding to each load level.

### 2.3. Loading Conditions

Two load levels, namely, 30% and 60% of the average ultimate flexural load, were used in the current study. The maximum normal tensile stresses applied on the specimen were 3.42 MPa and 6.84 MPa, corresponding to a load of 4.11 kN and 8.22 kN, respectively. Note that at these load levels, no obvious crack was observed, as the load level was lower than the first cracking strength. The apparatus for the sustained loading is shown in Figure 1. It consists of a load cell, three plates (30 mm in thickness), two threaded pillars (20 mm in diameter), and several cylinder steel blocks (20 mm in diameter), allowing us to load three specimens simultaneously [46]. The load was applied by adjusting the nuts, which were monitored by the load cell mounted between the top two cover plates. When the target load level was achieved, the load was sustained by adjusting the bolts every 24 h during the tests. To eliminate the influence of the gravity of the upper specimen on the lower specimen, the loading apparatus together with specimens were laid horizontally in the carbonation chamber. Note that side surfaces were sealed by epoxy before load. A specimen without loading (0% load level) was placed in the carbonation chamber as a reference. The surface exposed to carbonation was the same as the loaded specimens.

### 2.4. Accelerated Carbonation Test

The accelerated carbonation test was carried out in accordance with GB/T 50082-2009. The exposure environment maintained a CO_2_ concentration of 20%, relative humidity of 70% and temperature of 20 °C. The accelerated carbonation process was measured for 7, 14 and 28 days. The accelerated carbonation process under sustained load was measured for 7, 14 and 28 days. At the appointed time, the specimens were taken out and cut along planes A, B, C and D. These four planes were under 0, 0.25, 0.5 and 1 of the applied maximum bending moment in zone II, respectively. The two loading conditions and four positions made 8 test profiles. The stress level corresponding to each test position is shown in Table 3. The carbonation depth was measured at the side subjected to maximum tensile stress. Regarding the measurement, the freshly cut surface was treated with 1% phenolphthalein solution. In the carbonated area, the alkalinity was low (pH < 9) and no colour change occurred. On the other hand, the colour of the uncarbonated area turned to purple due to the alkaline nature of the bulk materials [47]. For each cross-section, carbonation depth was calculated by averaging depths at the middle of 10 equivalent subareas along the width. For each condition, three specimens were tested.

### 2.5. Microhardness Measurement

After accelerated carbonation, samples close to positions A and B and within zone I and zone II of the prismatic specimen were collected for the microhardness (Vickers hardness) measurement, respectively. Zone I corresponds to the area under no load and Zone II is under the maximum bending moment. The sample was cut from the exposure surface with a size of 20 mm × 20 mm × 20 mm. An HXD-1000TMC microhardness indenter (Shanghai Optical Instrument Factory, Shanghai, China) instrumented with a Vickers pyramid diamond indenter was used for the test. In total, 60 indentations were carried out along the depth perpendicular to the exposed surface. The distance between each adjacent indentation was 20 μm. The applied maximum load was 200 N and the dwell time was 10 s. As microhardness tests are very sensitive to the smoothness of the test surface, grinding and polishing were performed prior to the test [48].

### 2.6. MIP Measurement

To investigate the coupled influence of sustained loading and carbonation on the porosity and pore size distribution, a mercury intrusion porosimetry (MIP) test was conducted. The tested samples were approximately 2 g, with a maximum dimension of 2–3 mm. Samples were collected from the specimens under 60% of the ultimate load level after 28 days accelerated carbonation. As shown in Figure 2, samples L60-I-C and L60-I-U were cut from zone I, where the materials can be considered unloaded. L60-I-C was taken from a distance perpendicular to the exposed surface within 6 mm (within the carbonation depth), while sample L60-I-U was collocated from 6–12 mm beyond the carbonation depth. Samples L60-II-C and L60-II-U were taken from zone II where the maximum bending moment was placed. They were taken from a depth within 10 mm (<carbonation depth) and 10–15 mm (>carbonation depth), respectively. The collected samples were then immersed in isopropanol for 3 days. Afterwards, they were dried at 45 °C for 48 h using a vacuum. The MIP tests were conducted using an Autopore II 9220 mercury intrusion porosimeter. The adopted contact angle was 130°. The pressure capacity of the instrument is 4.00 × 10^−3^–4.13 × 10^2^ MPa.

### 2.7. TG/DTD and XRD Test

To test the change along the carbonation depth, TG/DTG and XRD were conducted on the samples taken from various depths of the specimens under coupled sustained loading (60% ultimate load) and accelerated carbonation for 28 days. Prior to the test, the hydration of the samples was stopped using the solvent exchange approach (isopropanol) [48]. The samples were stopped using solvent exchange approach (isopropanol) [48]. The TG/DTG test was performed using an SDT Q600 thermogravimetric analyzer (TA Instruments). The TG/DTG test was conducted using N_2_ gas with a flow rate of 100 mL/min and a heating rate of 10 °C/min, within a temperature interval between 20 °C and 1000 °C.

A Rigaku Miniflex 600 X-ray diffractometer was used for the XRD test. The adopted Cu-Ka radiation has a voltage of 40 kV and a current of 40 mA. The scanning speed was 0.02°/s in a range of 10–90°.

## 3. Results

### 3.1. Carbonation Depth

Figure 3 shows the measured carbonation depth under different loading conditions. In general, the carbonation depth increased with carbonation time. In terms of the unstressed sample, the carbonation depth was 2.3 mm after 7 d accelerated carbonation, and it reached 3.37 mm and 4.78 mm at 14 d and 28 d carbonation, respectively. Clearly, the carbonation rate decreased with time. This also applies for the specimens under sustained load. This can be attributed to the densification of the carbonated zone [32]. The redistribution of pore sizes is reported in Section 3.4 to support this argument.

When sustained load was applied, the carbonation depth was significantly increased. The higher the stress level, the larger the carbonation depth. This is in accordance with the plain concrete [49]. Furthermore, the stress level had a more remarkable influence on the carbonation depth in a short period than the longer-period exposure. At the flexural stress level *δ_T_* = 0.6 (S6-PD), the carbonation depths were 7.21 mm and 9.92 mm, respectively, after 7 d and 28 d carbonation. The carbonation depth was more than 3 times that of the unstressed condition at 7 d, and it decreased twofold at 28 d. In terms of S3-PB, the sample was under a stress level *δ_T_* of 0.075, and the carbonation depth increased by about 100% at 7 d. As the carbonation time went on, the increment of carbonation became smaller. The values were 53.1% and 41.2% for the 14 d and 28 d carbonated specimens, respectively. This is because, under sustained loading, short-term creep as well as microcracks occurred, which increased the porosity and the connectivity of the matrix. This sped up the ingress of CO_2_ into the matrix. On the other hand, as illustrated in Section 3.4, the carbonation generated CaCO_3_, which filled the small pores, and microcracks in the carbonated area became compact, meaning the channels for CO_2_ diffusion were somewhat blocked. This led to a reduction in carbonation rate.

In the case of S6-PC and S3-PD, they were under the same flexural stress levels, i.e., *δ_T_* = 0.3. The carbonation depth after 7 d carbonation was the same. With a prolonged carbonation period, the carbonation depth of S6-PC became larger than that of S3-PD. The same trend was observed for S6-PB and S3-PC. The possible explanation is as follows. As the samples taken from the specimen under 60% ultimate load were closer to the support, more shear stress occurred compared with the position with same flexural stress level in the specimen loaded with 30% of the ultimate load. The increased shear stress may generate microcracks or increase the porosity, which promotes CO_2_ ingression.

### 3.2. Carbonation Depth Model

In the case of accelerated carbonation, the carbonation depth was in proportion to the square root of exposure time [50]. The typical carbonation depth model proposed by Papadakis et al. [51] has been widely used, as shown in Equation (1):(1)$$x=\sqrt{\frac{2DC_{0}t}{m_{0}}}$$
where *x* is the carbonation depth of concrete (mm), *D* the effective diffusion coefficient (mm^2^/s), *C*_0_ the CO_2_ concentration at the concrete surface (mol/m^3^), *m*_0_ is the amount of CO_2_ that is completely absorbed by concrete (mol/m^3^) and *t* is the carbonation time (s). It is usually simplified as Equation (2) [52,53].
(2)$$\begin{array}{l} x=K\sqrt{t}\\ K=\sqrt{\frac{2DC_{0}}{m_{0}}} \end{array}$$
where *K* is the carbonation rate of concrete (mm/s 0.5). It is an important indicator of the resistance to carbonation, which is determined by both the exposure environment and concrete mix [49,54]. *D* represents the penetrability of CO_2_ into the concrete, which is mainly affected by the micropore structure of the matrix, as well as the temperature. *m*_0_ denotes the ability of concrete to absorb CO_2_. It is mainly determined by the mix proportions. As external load is applied, and especially a sustained load, the micropore structure changes, thus leading to a change in the penetrability of CO_2_. This influence can be considered by Equation (3) [55,56,57]:(3)$$\begin{array}{l} x_{\sigma }=K_{ks}K_{0}\sqrt{t}\\ K_{0}=\sqrt{\frac{2D_{0}C_{0}}{m_{0}}} \end{array}$$
where *x_σ_* denotes the carbonation depth under sustained load, *K_ks_* is the stress influence coefficient, *K*_0_ is the carbonation rate of concrete under no load, and *D*_0_ is the initial diffusion coefficient of CO_2_. Equation (3) can be further simplified as the following:(4)$$\begin{array}{l} x_{\sigma }=K_{ks}x_{0}\\ x_{0}=K_{0}\sqrt{t} \end{array}$$
where *x*_0_ denotes the carbonation depth under no stress. *K*_0_ was calculated as 0.869, 0.901 and 0.903 for 7, 14 and 28 d, respectively. The calculated stress influence coefficient *K_ks_* under various flexural tensile stress levels *δ_T_* and exposure times is shown in Table 4.

Figure 4 plots the change of *K_ks_* with the stress level *δ_T_.* Clearly, the linear relationship commonly observed for the conventional concrete [58] does not apply here. The increase rate of *K_ks_* decreases with the *δ_T_*. This can be attributed to the bridging effect of the fibers, which limits the change of the micropore structure under a sustained load. An exponential function as shown in Equation (5) was therefore proposed.
(5)$$K_{ks}=1+y_{a}\delta_{T}{}^{y_{b}}$$
here, *y*_a_ and *y*_b_ are the fitting parameters. The fitting results are shown in Figure 4. Clearly, a high determination coefficient (R^2^) is observed for all three exposure periods. The fitting parameters *y*_a_ and *y*_b_ change with the exposure period *t* and can be expressed by the following equations:(6)$$y_{a}=2.903-0.059t$$
(7)$$y_{b}=0.010\exp(t^{0.132})$$

The relationship between *K_ks_* and *δ_T_* can therefore be expressed as:(8)$$K_{ks}=1+(2.903-0.059t)\delta_{T}{}^{\left(0.010\exp(t^{0.132})\right)}$$

Substituting Equation (8) into Equation (3), a modified carbonation depth model that can be used to consider the coupled effect of flexural tensile stress and carbonation time was obtained, as shown in Equation (9).
(9)$$x_{\sigma ,t}=(1+(2.903-0.059t))\delta_{T}{}^{\left(0.010\exp(t^{0.132})\right)})K_{0}\sqrt{t}$$

The fitted curves for 7 days, 14 days and 28 days are shown in Figure 5. The determination coefficient corresponding to each carbonation period is above 0.9, confirming the agreement between the prediction and experiments.

### 3.3. Mineral Phases

For the TG/DTG measurement, specimens subjected to a coupled sustained 60% load level and accelerated carbonation for 28 days were used. Samples in Zone II (pure bending zone) at different depths (0−4 mm, 4−8 mm, 8−12 mm, 12−16 mm and 16−20 mm) perpendicular to the carbonation surface were tested. The TG/DTG test results are shown in Figure 6. The DTG is a derivative of the TG curve, which shows peaks with respect to various phases. The peak at 100−125 °C corresponds to the ettringite and C-S-H [59]. Clearly, the intensity decreases with the depth. The peak at around 240 °C may correspond to the decomposition of PVA fibres. The peak observed at about 450 °C can be attributed to the dehydration of calcium hydroxide (CH). It can be seen that in the non-carbonated part, calcium hydroxide exists. As the depth decreases, the amount of calcium hydroxide decreases gradually, and it disappears when the depth is smaller than 12 mm. This is in accordance with the observation in Section 3.1, in which the carbonation depth was determined using phenolphthalein. As can be seen from the figure, as the depth decreases, the peak of calcium hydroxide becomes weaker first. Once the calcium hydroxide is depleted, the intensity of ettringite and C-S-H starts to decrease. This confirms the carbonation sequence of the hydration products. As reported by [60,61], the mass change between 550 °C and 900 °C is assumed to be the mass of CO_2_ released from CaCO_3_. The calculated CO_2_ content along the penetration depth is shown in Figure 7. Three regimes can be distinguished. The first regime was at the depth of 0−8 mm with a high carbonation degree. In this regime, the sample within the depth of 0−4 mm absorbed about 16 wt. % of CO_2_. The absorbed CO_2_ gradually decreased to 15.4 wt. % at the carbonation depth of 4−8 mm. A significant drop (about 18%) can be observed in region II at a depth of 8−12 mm. This can be regarded as partly carbonated. Afterwards, the CO_2_ content decreased gradually, and seems to reach an asymptotic value when the depth was greater than 16 mm. This means that no carbonation occurred at this depth, which is greater than that measured by the phenolphthalein, i.e., 10 mm.

Figure 8 compares the XRD patterns of samples at depths of 0−4 mm and 16−20 mm, respectively, in Zone II. The samples were taken from a specimen under a coupled sustained 60% load level and accelerated carbonation for 28 days. As revealed by the TG/DTG, the area within 4 mm had a high degree of carbonation, and no carbonation can be expected when the depth is larger than 16 mm. Clearly, a strong diffraction peak of quartz was observed in both samples due to the presence of quartz sand. A much stronger intensity of calcite could be seen in the carbonation area, which was accompanied by the disappearance of portlandite and the reduction of ettringite. This confirms the observation reported by the TG/DTG.

### 3.4. Porosity and Pore Size Distribution

Figure 9 shows the MIP test results for samples at zone I and zone II of specimens after 28 d of coupled sustained 60% ultimate load and carbonation. The porosity values of the samples at depths of 0−6 mm (L60-I-C) and 6−12 mm (L60-I-U) under no stress were 8.70 and 11.57%, respectively. Although the critical pore diameter of L60-II-C was 88.38 nm and larger than that of L60-II-U (26.12 nm), it can be observed in Figure 10a that the porosity in the range of 0.1–10 μm was reduced more than 2-fold after carbonation. This can be explained by the fact that the reaction of gaseous atmospheric CO_2_ with the calcium-bearing phases densifies the micropore structure due to the formation of calcium carbonate [62]. In this process, pores in the size range of 0.1–10 μm are filled and turn into pores with sizes smaller than 0.1 μm. The mesopore (>10 μm) volume is reduced as well. More specifically, the mesopore decreases from 1.42% to 1.28% after 28 d of accelerated carbonation.

In the area under coupled sustained flexural stress and carbonation, the porosity values at 0–10 mm (L60-II-C) and 10–15 mm (L60-II-U) were 9.33% and 9.38%, respectively. The critical pore diameter changes were 62.59 nm and 77.68 nm for L60-II-U and L60-II-C. It should be noted that, although it is indicated by the phenolphthalein solution that L60-II-U was beyond the carbonation depth, the TG/DTG results show that carbonation partly occurred in this area. The resulting porosity is a combination of the sustained flexural stress and carbonation. Compared with L60-II-C, L60-II-U had a lower carbonation degree and flexural stress level. The two opposite effects led to the result that the two samples had almost the same porosity and critical pore diameter. However, due the sustained load, more pores with sizes larger than 10 μm were observed in L60-II-C. Simultaneously, the volume of pores smaller than 0.1 μm was reduced.

Compared with L60-I-C, the porosity of L60-II-C was increased as the volume of pores larger than 100 μm increased from 1.28% to 1.51%. However, much more pores in the size range of 0.1–10 μm turned into smaller pores (<0.1 μm) due to the increased carbonation degree. In terms of L60-II-U, as small flexural stress was applied, the total porosity was reduced compared with L60-I-U. This was accompanied with a reduction in mesopores larger than 0.1 μm and an increase in pores smaller than 0.1 μm. The measurements confirm that accelerated carbonation played an important role in refining the pores in the range of 0.1 to 10 μm, and the sustained load had a limited influence on generating pores (flaw) larger than 10 μm due to the bridging effect of the fibres [63]. The redistributed pore size distribution enhanced the resistance of microstructure to CO_2_ penetration.

### 3.5. Microhardness

Figure 11 shows the profile of microhardness perpendicular to the exposed surface. A reduction in microhardness is observed from the surface to the internal area in all cases. The microhardness of the carbonation area was significantly higher than that of the non-carbonized area, although the area closer to the exposed surface was under higher sustained flexural stress. This confirms that the carbonation process increased the mechanical properties of the matrix. This is in accordance with the results of [27,64], as in the carbonation area, the number of capillary pores larger than 100 μm corresponding to the mechanical properties [65] was significantly reduced. Similar results have been obtained by nanoindentation [66]. As the depth increased, the microhardness became a constant, and the inflection point could be used as an indicator of the carbonation depth. However, due to the heterogeneous nature of ECC, a large variation was observed in the current study. Therefore, one indentation is not statistically representative of the real condition at a certain depth. A larger dataset is required. Nevertheless, the general trend between carbonated area and non-carbonated area was captured. At the depth of 2 mm, the microhardness was in general greater than 100 HV, while this value decreased to the range of 60 to 80 HV. This is attributed to the redistribution of pore sizes, as reported in Section 3.3.

## 4. Conclusions

In the current study, the influence of sustained flexural load on the carbonation of ECC was investigated. A modified carbonation depth model that can be used to consider the coupled effect of flexural tensile stress and carbonation time was derived. The following conclusions can be drawn:(1)The stress level had a more remarkable influence on the carbonation depth within a short period of accelerated carbonation compared with a long period. At the flexural stress level *δ_T_* = 0.6 (S6-PD), the carbonation depths were 7.21 mm and 9.92 mm, respectively, after 7 d and 28 d carbonation. The carbonation depth was more than three times that in the unstressed condition at 7 d, and decreased twofold at 28 d;(2)In the carbonation depth model, an exponential relationship was observed between the stress influence coefficient and the flexural tensile stress level of ECC. This is different from the plain concrete, in which a linear relation is generally used. The determination coefficient corresponding to each carbonation period was above 0.9, confirming an agreement between the prediction and experiments;(3)Within the carbonation depth detected by the phenolphthalein solution, the change of absorbed CO_2_ content can be divided into three regimes, namely, highly carbonated, partly carbonated and uncarbonated. A sharp drop was observed in the partly carbonated area, while the content gradually decreased in the others;(4)Under the coupled sustained load and accelerated carbonation, the accelerated carbonation played an important role in refining the pores in the range of 0.1 to 10 μm. The sustained load mainly contributed to generating pores (flaws) larger than 10 μm. This effect was limited due to the presence of fibres;(5)Microhardness decreased from the exposed surface to the interior part. In general, areas with a greater carbonation degree showed greater microhardness, even under a large sustained load level, as the carbonation process refined the pore structure.

## Figures and Tables

**Figure 1 materials-15-06192-f001:**
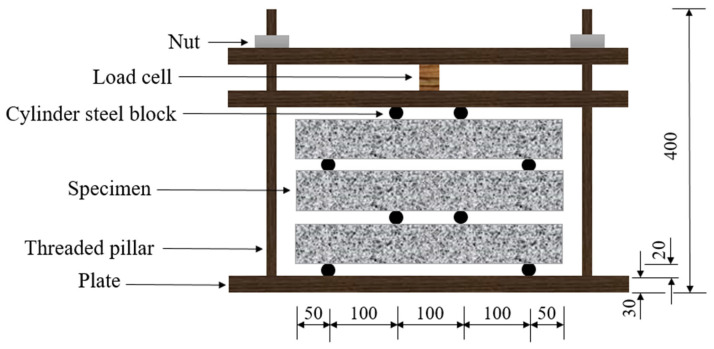
Schematic view of the apparatus for the sustained loading (unit: mm) (the full specimen is under accelerated carbonation).

**Figure 2 materials-15-06192-f002:**
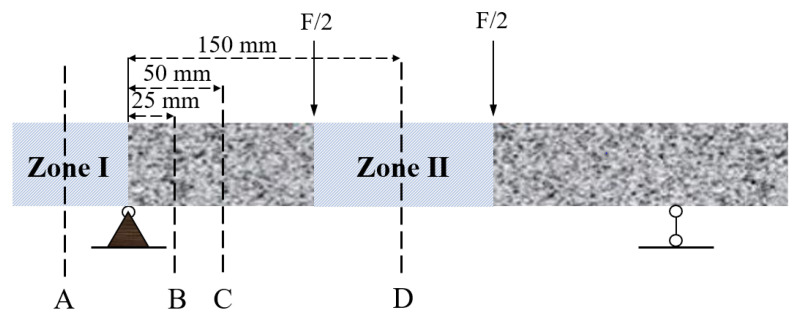
Schematic view of the carbonation depth measurement.

**Figure 3 materials-15-06192-f003:**
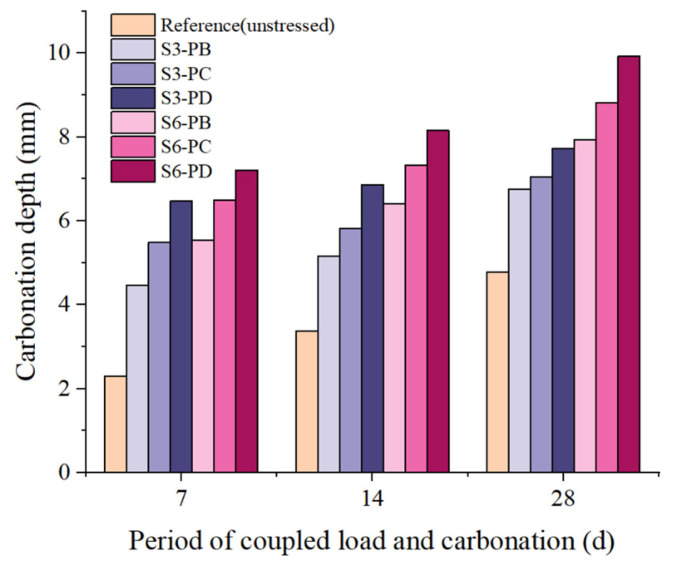
The measured carbonation depth of the area under different loading conditions.

**Figure 4 materials-15-06192-f004:**
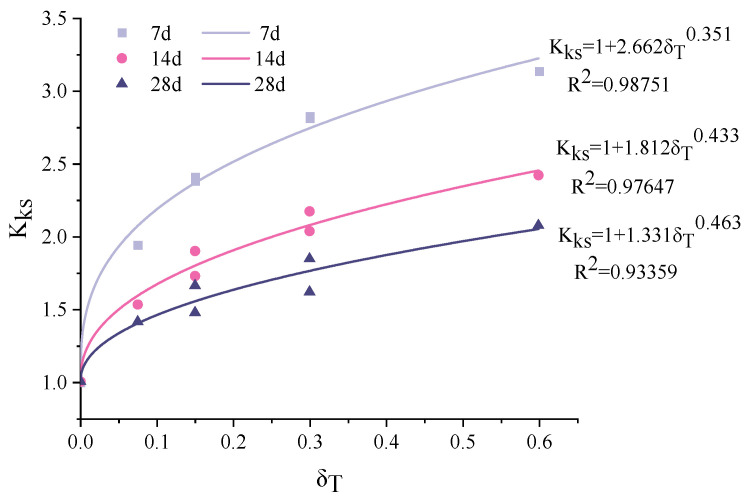
Relationship between stress influence coefficient and flexural tensile stress level.

**Figure 5 materials-15-06192-f005:**
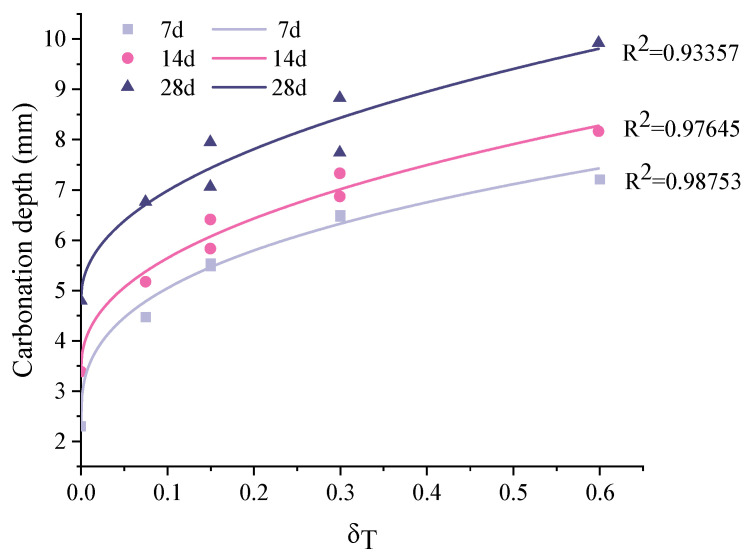
Comparison between the predicted carbonation depth and the experimental results.

**Figure 6 materials-15-06192-f006:**
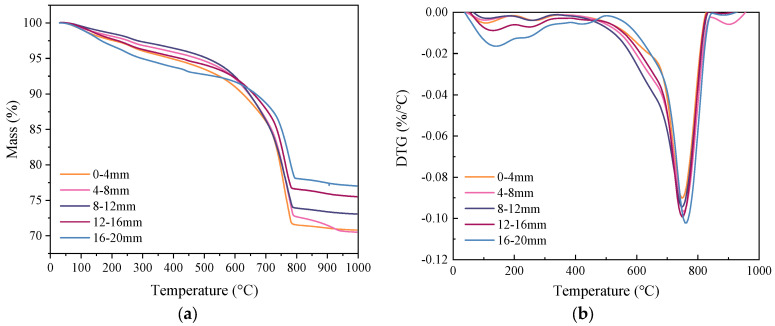
Results of (**a**) TG and (**b**) DTG at various depths.

**Figure 7 materials-15-06192-f007:**
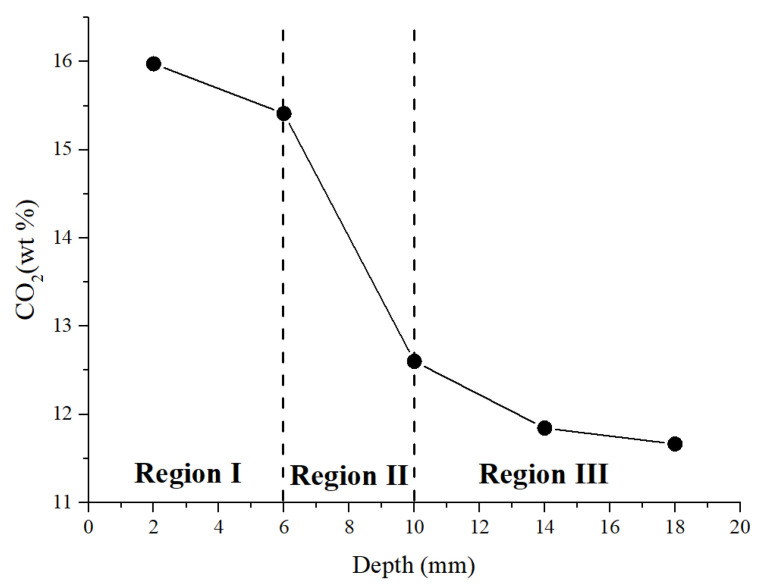
Change in CO_2_ content along the profile perpendicular to the exposed surface.

**Figure 8 materials-15-06192-f008:**
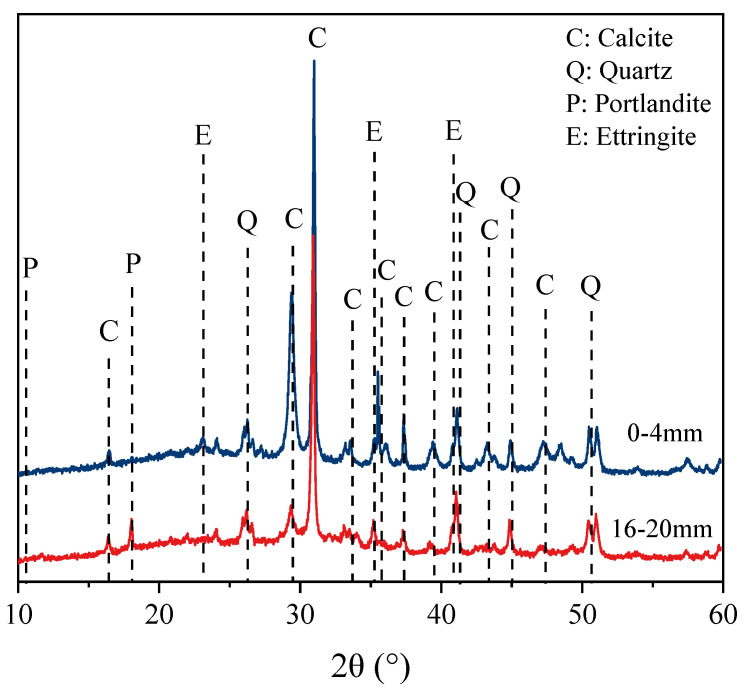
XRD results at different depths.

**Figure 9 materials-15-06192-f009:**
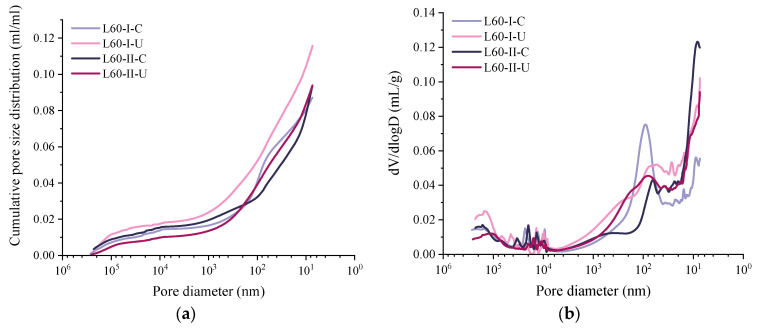
Pore size distribution of samples at different depths and load levels: (**a**) cumulative pore volume; (**b**) pore size distribution curves.

**Figure 10 materials-15-06192-f010:**
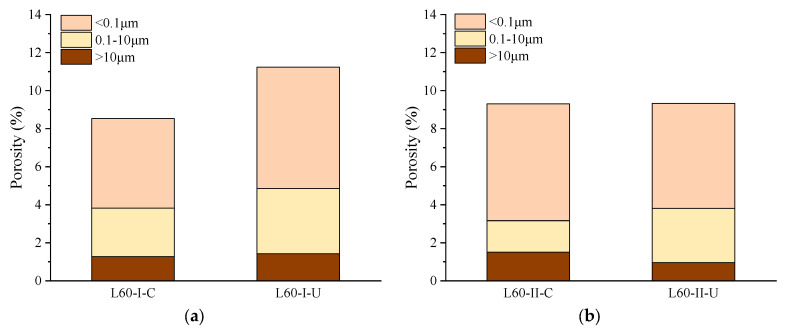
Comparison of the pore size distribution of samples at different depths and load levels: (**a**) zone I; (**b**) zone II.

**Figure 11 materials-15-06192-f011:**
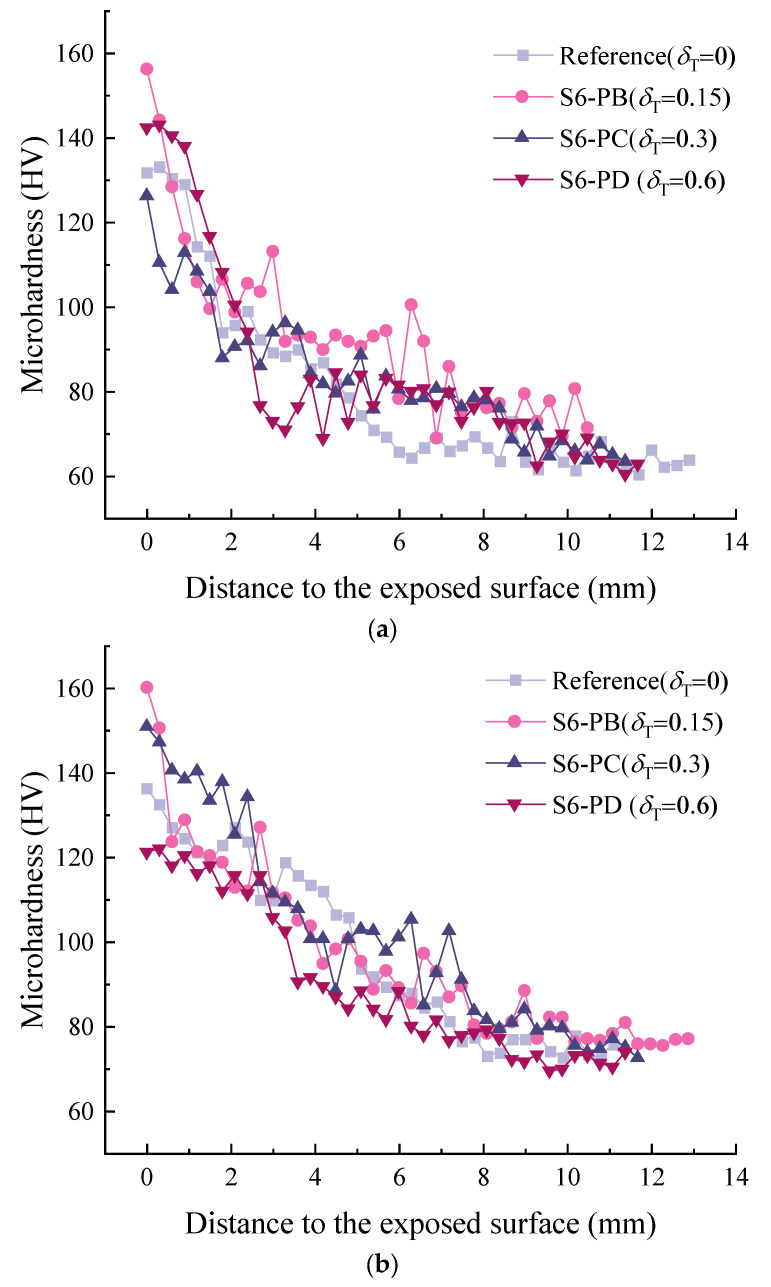
Change in microhardness along the profile perpendicular to the exposed surface after a certain carbonation period: (**a**) 7 d; (**b**) 14 d; (**c**) 28 d.

**Table 1 materials-15-06192-t001:** Mix proportions of ECC (kg/m^3^).

Cement	Fly Ash	Quartz Sand	Water	VMA	Fiber	W/b	Superplasticizer
563	676	450	372	4.50	26	0.30	5.58

**Table 2 materials-15-06192-t002:** Chemical compositions of the cement and the fly ash (wt. %).

Oxides	CaO	SiO_2_	Al_2_O_3_	Fe_2_O_3_	MgO	SO_3_	Na_2_O	K_2_O	TiO_2_	MnO	P_2_O_5_
Cement	63.21	18.48	6.74	3.45	3.24	3.16	0.17	0.53	0.35	0.27	0.16
Fly ash	3.43	49.66	35.97	5.77	0.63	1.12	0.62	0.93	0.99	0.04	0.28

**Table 3 materials-15-06192-t003:** Stress level *δ_T_* of the tested surface under different loading conditions.

Name	Load Level of Specimen	Position	δ*_T_*
Reference	0	A	0.000
S3-PB	0.3	B	0.075
S3-PC	0.3	C	0.150
S3-PD	0.3	D	0.300
Reference	0.6	A	0.000
S6-PB	0.6	B	0.150
S6-PC	0.6	C	0.300
S6-PD	0.6	D	0.600

**Table 4 materials-15-06192-t004:** Carbonation depths of specimens under different flexural tensile stress levels.

Load Level of Specimen	Position	δ*_T_*	Carbonation Depth/mm					
			7 d	*K_ks_*	14 d	*K_ks_*	28 d	*K_ks_*
0	C	0.000	2.300	1.000	3.370	1.000	4.780	1.000
0.3	A	0.075	4.470	1.943	5.160	1.531	6.750	1.412
0.3	B	0.150	5.480	2.382	5.820	1.727	7.050	1.475
0.3	C	0.300	6.470	2.813	6.860	2.036	7.730	1.617
0.6	A	0.150	5.540	2.409	6.400	1.899	7.940	1.661
0.6	B	0.300	6.500	2.826	7.320	2.172	8.820	1.845
0.6	C	0.600	7.210	3.135	8.160	2.421	9.920	2.075

## Data Availability

The data presented in this study are available upon request from the corresponding author.

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
