# Peer review of "Carbonation Behavior of Engineered Cementitious Composites under Coupled Sustained Flexural Load and Accelerated Carbonation"

_materials, 2022, doi:10.3390/ma15186192_

Round 1
Reviewer 1 Report
Abstract: The results are not clear.
1. Introduction:
Line 59-60: The statement is not clear.
Line 68-72: The sentences are same with the Line 72-76.
In Introduction section, it should also explain the reasons why considering "the sustained flexural load". Because in fact, if the structural concrete is damaged, it is very likely that the related structural elements will failure due to the shear.
In addition, it is also necessary to explain why use "accelerated carbonation at 7, 14 and 28 days". It should be explained why not use the effect of carbonation at 1, 2, 3 months or more.
2. Materials and Methods
Line 98: Please state the number of beam specimens used.
2.2. Line 113: for "the ultimate flexural strength of 11.41 MPa", also state the value of the ultimate flexural load in kN units.
2.3. Line 127-128: The statement is not clear.
In Figure 1: Please draw clearly the "Carbonation Zone (I & II)" on the beams under study, whether only on the bottom beam or all beams.
2.4. Please describe what is meant by "accelerated carbonation test"
Line 136-137: The statement is not clear.
Based on Figure 1: Please explain whether all the beams are marked by "Zone I & II" or only on the lower beams? If you only look at the carbonation effect on the lower beam, so what is the function of the beams above it?
Please explain also whether the cutting of the A, B, C & D planes is done after the flexural loading is completed at the age of 7, 14 & 28 days or what?
2.5. Line 156: Actually the shear force from the support and the flexural force affect the stress flows at Zone I.
3. Results
3.1. Carbonation graph.
Based on Figure 3: PD samples always exhibit higher carbonation depth compared to PC & PB samples. Even though PD samples are in Zone II which uses an epoxy coating on both sides. Please explain.
Based on Figure 7 (b): It is important to explain why after temperature of 750 0C, the DTG value goes to 0.
4. Conclusion: Please provide more specific conclusions, and show values that are in accordance with the test results.

Reviewer 2 Report
The submitted article (Carbonation behavior of engineered cementitious composites under coupled sustained flexural load and accelerated carbonation) investigates the behaviour of engineered cementitious composites under coupled sustained flexural load and accelerated carbonation. This topic is important for the construction industry and could be considered for publication in the MDPI journal of Materials. However, the following issues must be addressed.
1- Improve the English language; there are grammar and spelling mistakes
2- The abstract should be improved. You need to mention what chemicals you used and at percentages. Remember the abstract stands alone, so it must be rich in information (brief).
3- Fly ash is widely used in the literature. Thus, you need to highlight the novelty of your work.
4- The Materials and Methods section is not well supported by references. For example, why did you use the grain size in the range 125-180 μm
5- I think Fig 3 must be a bar chart, not curves or lines, because 3 points are not enough to generate a reliable line or curve
6- Mention the references of all used equations, such as Eq.2
7- Avoid mass citation, do not use more than 2-3 references per piece of information. For example, you used [29, 44-47], and you used 3 references to support one equation
8-It would be better to mention the recent relevant literature, such as the following study:
https://doi.org/10.1016/j.conbuildmat.2018.08.021
Round 2
Reviewer 1 Report
No comments